# At the Edge of Orthopaedics: Initial Experience with Transarterial Periarticular Embolization for Knee Osteoarthritis in a Romanian Population

**DOI:** 10.3390/jcm11216573

**Published:** 2022-11-05

**Authors:** Octav Russu, Florin Bloj, Andrei Marian Feier, Vlad Vunvulea, Stefan Mogoș, Vlad Predescu, Tudor Sorin Pop

**Affiliations:** 1Department of Orthopaedics and Traumatology, George Emil Palade University of Medicine, Pharmacy, Science, and Technology of Targu Mures, 540142 Targu Mures, Romania; 2Ares Excellence Center, Monza Oncological Hospital, 013812 Bucharest, Romania; 3Doctoral School, George Emil Palade University of Medicine, Pharmacy, Science, and Technology of Targu Mures, 540142 Targu Mures, Romania; 4Department of Anatomy and Embriology, George Emil Palade University of Medicine, Pharmacy, Science, and Technology of Targu Mures, 540142 Targu Mures, Romania; 5Department of Orthopedics and Traumatology, Foişor Orthopaedics Hospital, 030167 Bucharest, Romania; 6Ponderas Academic Hospital, 021659 Bucharest, Romania; 7Department of Orthopedics and Traumatology, St Pantelimon Clinical Hospital Bucharest, University of Medicine and Pharmacy ‘Carol Davila’, 050474 Bucharest, Romania

**Keywords:** osteoarthritis conservative treatment, endovascular embolization, periarticular embolization, transarterial osteoarthritis embolotherapy

## Abstract

Background: Transarterial embolization (TAE) of genicular artery branches is a relatively new technique that has emerged as a promising method for delaying invasive knee surgery in patients suffering from degenerative knee osteoarthritis (OA). In mild to moderate OA, invasive major surgery can be safely postponed, and patients with major risk factors now have an alternative. Our aim was to examine the impact of TAE on clinical outcomes in individuals with degenerative knee OA over a 12-month period. Methods: A case series of 17 patients diagnosed with knee OA and treated with TAE was included in the study. Every patient was clinically evaluated at different timeframes according to the Western Ontario and McMaster Universities’ arthritis index, knee injury, and osteoarthritis outcome scores, and the 36-item short-form survey (WOMAC, KOOS, and SF-36). Results: At the first follow-up (1 month), KOOS and WOMAC improved from 46.6 ± 13.2 (range 27.3–78.2) to 56.5 ± 13.9 (range 32.3–78.4; *p* = 0.023) and 49.5 ± 13.2 (range 29.3–82.3) to 59.8 ± 12.6 (range 39.3–83.5, *p* = 0.018), respectively. Physical SF-36 improved significantly from 42.1 ± 7.75 (range 30.3–57.3) to 50.5 ± 9.9 (range 35.6–67.9; *p* = 0.032). No significant changes in scores were observed at three, six, or twelve months after TAE. Conclusions: TAE provided early pain reduction and considerable improvement in quality of life without complications for a consecutive sample of Romanian patients with mild to severe knee OA.

## 1. Introduction

Osteoarthritis (OA) is the most common form of inflammatory and degenerative joint disease, affecting more than 40 million people across Europe [1]. A drastic increase in OA incidence is expected as underdeveloped Eastern European countries gain access to basic medical screening and treatment options. The knee is the most affected anatomical site [2], and total knee replacement (TKR) is the main invasive curative treatment as of today [3]. TKR indications and patient selection are now standardized [4], and an increased number of patients are poor candidates for the procedure. Frequent comorbidities such as obesity, diabetes, dementia, cirrhosis, and immunodeficiencies are shown to increase the TKR complication rate [5]. Morbidly obese patients scored lower on subjective tests at a 10-year follow-up than non-obese or obese patients [6]. In patients with dementia, the invasive procedures might aggravate the base condition and increase resource utilization [7]. Furthermore, a recent nationwide cohort study proved the increase risk of OA advancement in patients with dementia [8]. Smokers are a population group that have been recently categorized at increased risk of surgical complications after TKR; they have a higher postoperative incidence of overall surgical complications, pneumonia, and revision surgery [9]. Due to increased risks of complications, an alternative, less invasive procedure for patients in these population groups is continuously sought [10].

A key factor in selecting an optimal patient for TKR is chronological age. Due to demanding physical activities and increased mechanical loading, young patients that undergo TKR are prone to increased revision rates and implant aseptic loosening in the first decade after surgery [11]. The current state-of-the-art does not assess TAE in young patients with different etiologies and mild to moderate knee OA. Highlighting the importance of such prospective outcome-based analyses is needed. On the other hand, senior candidates are likely to acquire postoperative infections and are at an increased risk of periprosthetic fractures. Even if the selection criteria are appropriately applied and the surgical technique is performed optimally, a relatively high proportion of patients do not report improved quality of life or pain relief at final follow-ups [12].

First-line conservative treatment consists of lifestyle behavioral changes in combination with physical therapy, NSAIDs, oral and topical analgesics, and eventually intra-articular infiltrations with viscoelastic or corticosteroid constituents. Currently, patients that have undergone the standard protocol of conservative knee OA treatments and exhausted all alternatives have no other viable option at hand. Moreover, with repeated follow-ups and increased subjective pain, the tendency of increasing analgesics and anti-inflammatory dosages and strength has been reported [13]. Gastrointestinal and vascular complications related to NSAIDs’ use and over prescription can now be predicted and decisions can be made [14].

In moderate stages of OA, synovial inflammation due to degenerative processes around the knee stimulates angiogenesis [15]. A hypothesis implies that OA chronic pain or residual pain after TKA is the result of a combination of hypervascularization and angiogenesis processes of synovial tissue and the joint capsule. Secondary to this, an increase in the number of local pain receptors also contributes to bone structural damage and chronic pain [16]. A minimally invasive procedure used by interventional radiologists for bleed control or tumor vascularization block is fluoroscopically-guided transcatheter arterial embolization [17]. A new strategy was implemented for patients with symptomatic knee OA: the transarterial embolization (TAE) of genicular arteries with a brachial approach. By selective catheterization and embolization of specific arterial genicular branches, vascular supply is blocked, the release of pro-inflammatory mediators is decreased, and nociceptive triggering is diminished [17]. As an alternative to invasive procedures, the curative effect of the procedure is not yet proven [18]. However, pain management and an increase in quality of life was previously shown to aid patients with contraindication to TKR.

In the current work, the clinical follow-up outcomes of patients that underwent embolization with TAE for symptomatic knee OA were evaluated. The aim was to demonstrate the safety and efficacy of the procedure in a successive case series at different follow-ups.

## 2. Materials and Methods

### 2.1. Design

The study was conducted according to the principles of the Helsinki Declaration. A local ethical committee’s approval was obtained. All patients signed the informed consent and agreed to have their imaging included in the study. At the time of enrollment, the patients agreed not to continue any intra-articular therapy for the ongoing follow-up time. A total of 17 patients were included in this case series report. Patient group structuring based on the Ahlbäck stage of OA can be seen in Figure 1.

### 2.2. Patient Selection

Patient selection was completed by the main author (an interventional radiologist) in collaboration with a senior orthopedic surgeon. A successive series of cases that met the inclusion criteria were nominated to be included. Inclusion criteria were as follows: symptomatic degenerative knee OA (Ahlbäck stage II or III), contraindication (of any reason) to TKR, >50 years of age, previous conservative treatment (of any type), and written informed consent and willingness to be included in an innovative treatment approach for knee OA. Exclusion criteria were as follows: patients that underwent any type of invasive surgery to the respective knee; stage IV or stage I Ahlbäck knee OA; and a history of renal failure or insufficiency.

### 2.3. Subjective Clinical Assessment

The knee injury and osteoarthritis outcome score (KOOS) [19] was used to analyze subjective outcomes. Patients were also asked to fill out the Western Ontario and McMaster Universities’ arthritis index (WOMAC) [20] at every follow-up visit. KOOS is an extension of WOMAC, but differences between them have been reported [21]; therefore, the methodology included both scores. Quality of life at different timeframes was assessed using the 36-item short-form survey (SF-36) [22]. Before undergoing the TAE procedure, each individual filled out KOOS, WOMAC, and SF-36 questionnaires. Follow-ups were performed at one, three, six, and twelve months after TAE intervention. All patients were elderly, so all submissions were made under the supervision of a study nurse. All the questionnaires were self-administered, and their language was Romanian. Demographic data were collected at the time of enrollment.

### 2.4. Imaging Evaluation

The knee OA Ahlbäck grading was assessed by the senior orthopedic surgeon using plain antero-posterior and lateral view radiographs at the time of physical examination. Magnetic resonance imaging (MRI) evaluation was performed for each patient in order to accurately identify arterial branches responsible for capsular inflammation (as seen in Figure 2).

### 2.5. Transarterial Periarticular Embolization: Step by Step

The procedure begins with the catheterization of the left brachial artery and the placing of a 5Fr catheter at the level of the superficial femoral artery. Afterwards, a selective catheterization of the superior genicular arteries (medial or lateral) is performed by using a 0.021 inch microcatheter (Direxion, Boston Scientific, Marlborough, MA, USA). Under fluoroscopic guidance, the desired genicular artery branch is identified. In total, 200 mcg of nitroglycerine is injected, and the knee is immediately iced for 10 min by using an instant cold pack of ice (Dynarex, Blauvelt, NY, USA). Then, an angiogram reveals hypervascularity with numerous capillary branches within the medial or lateral joint space, as exemplified in Figure 3a,c. A suspension with contrast substance and imipenem/cilastatin sodium (IPM/CS) (embolic agent) was injected. Distal hypervascularity was assessed fluoroscopically until it was resolved. As seen in Figure 3b,d, adequate blood flow should be maintained within the selected artery branch after deployment. A control angiogram is performed as a final step, and hemostasis is achieved by manual compression of the brachial artery. All patients were scheduled to be discharged 4 h postintervention.

### 2.6. Statistical Analysis

Descriptive statistics were calculated for KOOS, WOMAC, and SF-36 at each follow-up visit. Demographic data were calculated as means and ± standard deviation. Clinical scores were evaluated at baseline and post procedure at each follow-up visit (1, 3, 6, and 12 months). Statistical comparisons were calculated for each follow-up related to the previous one and for each visit compared to the respective baseline score. The level of confidence was established at 95%. Based on its robustness, the Kolmogorov–Smirnov test was used to assess the normality assumption of data. A paired Student’s *t*-test was used, and *p* < 0.05 was set as the threshold for statistical significance. Microsoft Excel (v. 16.64) was used for calculating descriptive statistics and GraphPad Prism 9 (v. 9.4.1) for statistical test assessment.

## 3. Results

There were no technique or procedural complications. One patient requested an overnight hospital stay due to anxiety distress. Demographic data along with patient characteristics are presented in Table 1. There were no differences in outcomes based on any of the demographic or patient characteristics data.

### 3.1. Knee Injury and Osteoarthritis Outcome Score

As seen in Table 2, there is a statistically significant increase in KOOS at one month (*p* = 0.0082) after TAE. No major changes were seen comparing the first and three-month follow-up outcomes (*p* = 0.1). A slight decrease in KOOS was seen at 12 months compared to the first follow-up but without statistical significance (*p* = 0.0112).

### 3.2. Western Ontario and McMaster Universities Arthritis Index

Table 3 summarizes the WOMAC mean scores at each follow-up. There is a significant change in the WOMAC score at 1 month follow-up compared to the baseline (*p* = 0.001). No significant change in outcomes was seen at further follow-ups.

### 3.3. 36-Item Short-Form Survey

SF-36 outcomes were split into their respective mental and physical subscales. There was a significant increase in both physical functioning (*p* = 0.00005) and mental (*p* = 0.0058) subscales of SF-36 at one-month post procedure. As shown in Figure 4, the scores thereafter slightly decline at each visit but without statistical significance between final follow-ups (physical 6 months vs. 12 months, *p* = 0.1666; mental 6 months vs. 12 months, *p* = 0.850).

## 4. Discussions

The most notable finding of our case series analysis was that TAE improves functional outcomes one month after the procedure without substantial gradual improvement at three, six, or twelve months. Subsequent follow-ups did not demonstrate any improvement in terms of mental health compared to the prior evaluation, although they were significantly enhanced from the baseline. It has been previously proven that orthopedic patients commonly score lower in SF-36 physical subscale outcomes [23]. However, in our analysis, the physical subscale showed promising results at each follow-up compared to the preprocedural state.

Although TKR is an excellent treatment for advanced knee OA, it is still difficult to provide effective conservative therapy for patients who have contraindications to surgery or opt to delay an invasive procedure. The American Academy of Orthopaedic Surgeons lists physiotherapy, weight reduction, and anti-inflammatory medicines as conservative treatment options for OA [24]. The latter is currently still contradictory due to safety and efficacy questioning after TKR, with clear vigilances to be tallied when prescribed [25,26]. Intra-articular corticosteroid or hyaluronic-acid injections are further pain-relieving strategies, albeit their efficacy is still up for discussion in current research [27,28]. As these methods may hasten OA’s degenerative processes, their safety has recently come into question. Intra-articular injected corticosteroids have a short-term effect on mild to moderate knee OA, with the only effect recently proven to be a delay in core invasive treatment [29]. An option that appears to be safer is injectable hyaluronic acid. However, a recent meta-analysis by Jevsevar et al. revealed that viscosupplementation in OA has no meaningful impact on functional outcomes [30]. In a large cohort study that involved more than 50,000 patients that underwent TKR, a knee injection (with either corticosteroids or hyaluronic acid) 3 months prior to surgery was proven to lead to a 3% overall rate of postoperative periprosthetic infection [31]. Due to skin bacterial load and the use of improper techniques in these procedures, the aim for an alternative that eliminates the risk of septic arthritis is mandatory [32].

Medical insurance limitations, especially in Eastern European countries, combined with patient resistance to the substantial time commitments inherent in physical therapy are major setbacks in TKR. When taken together, these issues highlight the critical need for new therapeutic approaches. Several prospective trials [33,34,35,36] have shown that TAE is a significantly safer and more effective alternative. TAE has several known benefits, such as reduced risk of procedure complications, rapid symptom relief, and a low cost of treatment.

For almost a decade, interventional radiologists have employed the combination of imipenem/cilastatin sodium (IPM/CS) with an iodinated contrast material as an embolic agent, with varying degrees of success [37]. Yamada et al. conducted an in vivo experiment demonstrating that IPM/CS employed as an embolic agent is ineffective in occluding major vessels (renal arteries in rats) due to its relatively small particle size, around 40µm [38]. However, the combination of IPM/CS as an embolic agent in smaller arteries has been shown to be effective, and this combination is employed in musculoskeletal embolotherapy or arterial embolization in gastrointestinal hemorrhage caused by tumors [39].

Comparing IPM/CS with another constituent used for embolization (Embozene TANDEM™), Okuno et al. found that IPM/CS produced superior outcomes regarding WOMAC functional and pain outcomes, with substantial increases from baseline levels [36]. However, similar to our case series, patients were allowed to keep using synergistic conservative treatments, which could have biased end results. In a different study conducted by the same group of researchers, long-term (up to 4 years) improvement in knee function and pain symptomatology after TAE was demonstrated, with synovitis significantly reduced on a 24-month follow-up MRI [35]. Subcutaneous hemorrhage at the entry site has been a frequent complication in their investigation. In our analysis, by using a retrograde artery approach this issue has been fully averted. In a more recent and larger cohort analysis, Little et al. showed that the majority of patients noticed a significant improvement in their pain and functional status after TAE, which persisted for more than 12 months [40]. Only KOOS was used as a subjective analysis tool in their prospective study, but a whole-organ magnetic resonance imaging score was also assessed. They concluded that with a limited sample of patients, these procedures might be exposed to placebo-biased results and their importance should be taken in consideration. These results are synergistic with our current case series outcomes. For our investigation, every follow-up subjective score differed in terms of statistical significance when compared to the baseline scores. This was furthermore confirmed by the significant increase in the SF-36 physical function subscale at each follow-up. The perception-based SF-36 mental subscale reached almost pre-procedural scores at 6 and 12 months of follow-up. This further confirms the hypotheses of lacking correlation between the two subscales of the instrument [41].

In the current literature, a large number of patients reported experiencing discomfort throughout the procedure, a result that contradicts our findings [35,36]. Lack of pain could be due to the retrograde approach that we used. The patients’ discomfort was caused by the injection of contrast/embolic under pressure into the selected genicular arteries [36]. However, this pain subsided promptly after the procedure. Typically, an ice pack is put on the surface of the knee corresponding to the region to be embolized. Subsequent cold-induced vasoconstriction aids in minimizing imprecise embolization of bone-blood-supply arterioles. By administering a small dose of nitroglycerine beforehand, unintended embolization of muscle branches is also avoided. Qualitative tools such as the visual analogue scale could have been used when assessing pain during procedural steps, allowing for the clear measurement of patients’ perceived impact in future studies.

When introducing new alternative treatment approaches, cost-effectiveness indicators are brought up for discussion [37]. Taking into consideration the fact that all procedures were performed utilizing a radial approach, patients were able to be discharged from the hospital within four hours following intervention. The use of IPM/CS as the embolic agent instead of particle embolization also contributes to minimizing the procedure’s overall cost. Moreover, the femoral access site imposes a timeframe of bed rest, and the associated complications risk is not to be overlooked. By using a radial approach, hospitalization time, bed rest complications, and immediate patient reported quality of life is improved. Added to that, a radial approach enables the use of bilateral embolization through a single-entry vascular point. In comparison, a femoral approach would require bed rest, thorough hemostasis and prolonged time between procedures, if it were planned to be performed bilaterally.

Our findings pave the way for cutting-edge research into less invasive therapy opportunities for patients with knee OA who are opting to delay surgery. Indications, complication management, and public health effects may all benefit from further research, particularly if a wide range of treatment choices available could be compared. Improved patient outcomes and lower healthcare costs could result from decreasing the need for invasive surgeries, which carry the risk of potentially fatal complications.

### Limitations

The modest sample size was undoubtedly one of our study’s drawbacks, although full questionnaire completions were collected without losing any subjects from later follow-ups. The period could have been extended, and control MRI angiography highlighting the absence of hypervascularization over a longer timeframe may have strengthened our results. A recently reported complication of the procedure is asymptomatic bone infarcts, and they could have been pointed out on a simple follow-up MRI [42]. As the follow-up period was considerably long, tracking the type of drug ingredients used for prior analgesia and anti-inflammatory effects could have added value to our analysis. Another significant drawback was the absence of a control group and group stratification by OA grade.

## 5. Conclusions

This study demonstrated that TAE with IPM/CS as an embolic agent provided subjective pain relief and a significant increase in quality of life in patients with mild to severe knee OA without intra- or post-procedural complications. An in-depth comparative analysis is necessary as a further research outline to accurately address cost-effectiveness in comparison to other techniques or the various products utilized.

## Figures and Tables

**Figure 1 jcm-11-06573-f001:**
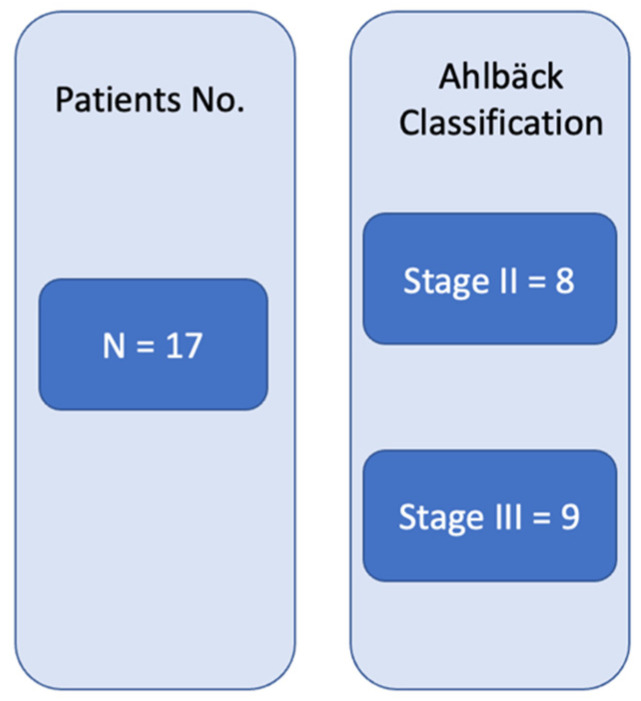
Patients’ OA radiographic classification.

**Figure 2 jcm-11-06573-f002:**
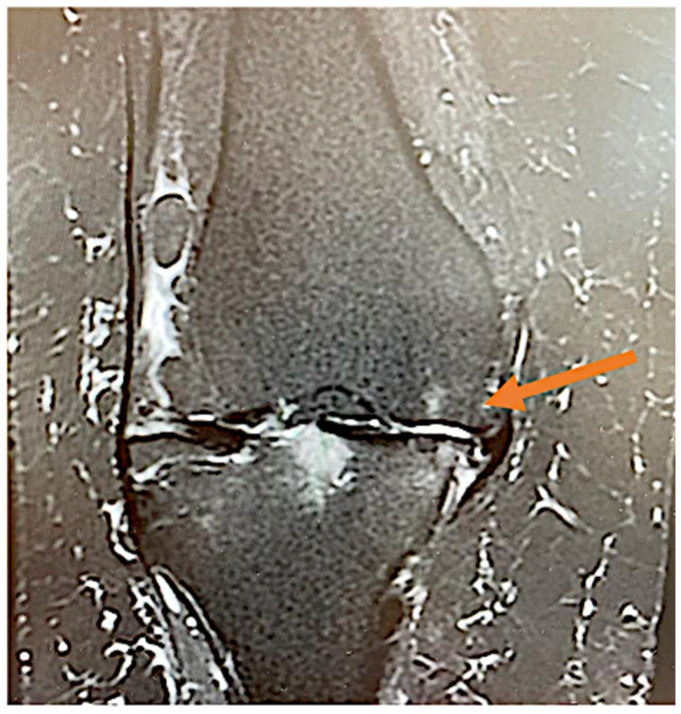
Knee coronary MRI STIR sequence highlighting (arrow), a neovascular arterial branch presumably responsible for knee pain in OA.

**Figure 3 jcm-11-06573-f003:**
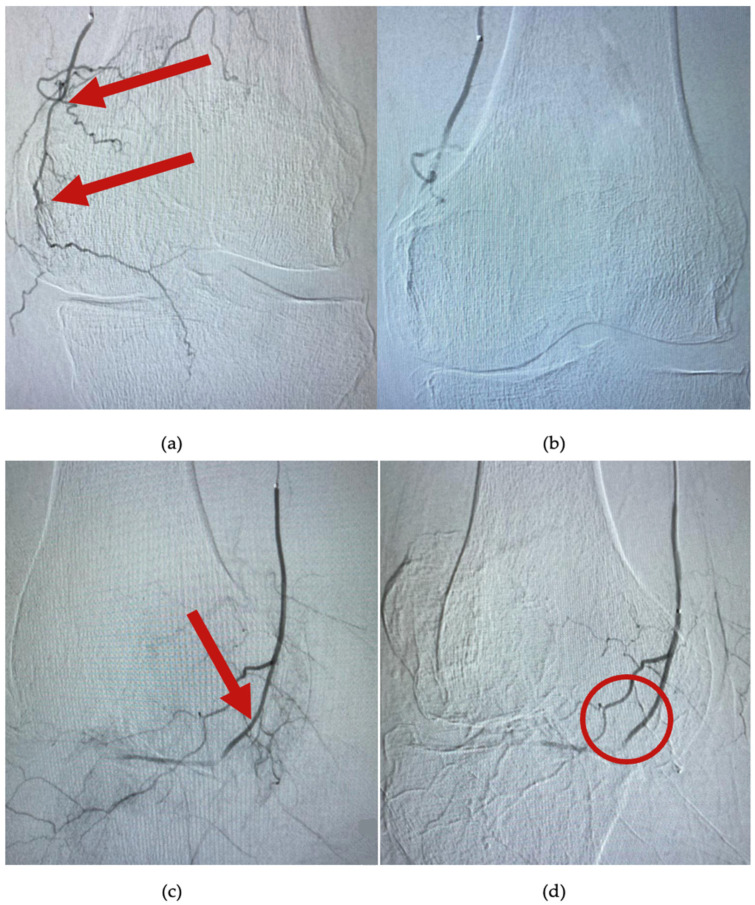
Angiogram at the level of the superior medial genicular artery. (**a**,**c**) are images of arterial blush (arrows), before embolization; (**b**) shows complete embolization and lack of the hypervascularity; and in image (**d**), bone and capsular branches are selectively avoided (circle).

**Figure 4 jcm-11-06573-f004:**
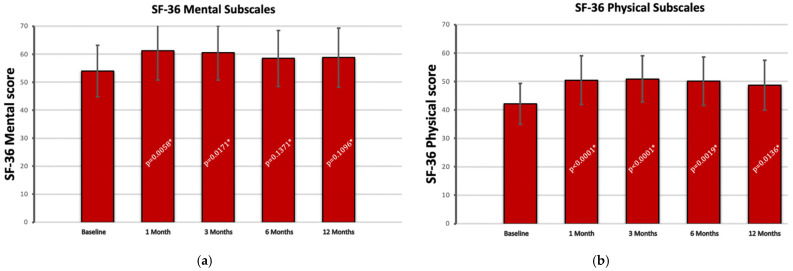
36-item short-form survey mental (**a**) and physical (**b**) subscales at each follow-up. * *p* values calculated compared to baseline scores.

**Table 1 jcm-11-06573-t001:** Patient demographics and characteristics.

Age (years), mean ± SD	64.4 ± 5.6
Height (cm), mean ± SD	163 ± 7.6
Weight (kg), mean ± SD	86.3 ± 14.3
BMI, mean ± SD	28.1 ± 4.2
Gender, male/female	5/12
Smokers, %, *n*	7
Daily physical activity, % (*n*)	17 (3)
Previous opioids usage, % (*n*)	41 (7)
Previous NSAIDs usage, % (*n*)	100 (17)
Physiotherapy, % (*n*)	47 (8)
Intra-articular injections, % (*n*)	29 (5)

**Table 2 jcm-11-06573-t002:** Knee injury and osteoarthritis outcome scores at different follow-up times (total).

Timeframe	Mean Value ± SD	*p* Value *	*p* Value **
Baseline	46.6 ± 13.2	n.a.	n.a.
1 month	56.5 ± 13.9	0.0082	0.0082
3 months	57.1 ± 14.1	0.1	0.0064
6 months	56.6 ± 13.4	0.2791	0.0070
12 months	56.0 ± 13.2	0.3089	0.0112

* Compared to previous follow-up. ** Compared to baseline. n.a. not available.

**Table 3 jcm-11-06573-t003:** Western Ontario and McMaster Universities’ arthritis index at different follow-up times (total).

Timeframe	Mean Value ± SD	*p* Value *	*p* Value **
Baseline	49.5 ± 13.2	n.a.	n.a.
1 month	59.8 ± 12.6	0.0017	0.0017
3 months	59.9 ± 12.1	0.9399	0.0014
6 months	59.8 ± 12.4	0.9387	0.0014
12 months	59.1 ± 11.5	0.3063	0.0116

* Compared to previous follow-up. ** Compared to baseline. n.a. not available.

## Data Availability

Not applicable.

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
