# Peer review of "At the Edge of Orthopaedics: Initial Experience with Transarterial Periarticular Embolization for Knee Osteoarthritis in a Romanian Population"

_jcm, 2022, doi:10.3390/jcm11216573_

Round 1
Reviewer 1 Report
In this study the Authors aimed to investigate the impact of transarterial periarticular embolization (TAPE) on clinical outcomes in patients with knee OA over a 12-month period.
There are some issues to be addressed and my specific comments are as follows:
- Materials and methods
a) Previous therapies have been indicated in Figure 1. Therefore, it is assumed that at the time of TAPE, patients were not undergoing any other types of treatment. It's correct? If so, the Authors should specify how long therapy has been discontinued.
b) In figure 1, treatment information appears to be misleading. The sum of patients in the three treatment groups (N= 21: 13+5+3) exceeds the total number of patients included in the case series (N=17).Furthermore, the information indicated in Figure 1 are not line with those included in Table 1. In this table, the authors reported that N=7 patients were previously treated with opioids, N=17 with NSAIDs and N=5 with intra articular injections. These is not in line with the information indicated in Figure 1 (NSAID treated total patients: N= 18; opioid treated total patients: N=3; IAHA treated total patients: N=8).
The Author should better clarify these points.
c) The Authors used parametric tests. Was the assumption of normality verified? The Authors should justify the chosen test and should add these specifications in the statistical analysis paragraph
- Results and Discussion
a) p-value comparison between baseline scores and those obtained at each time -point of follow up should be reported.
b) it should be interesting to evaluate if TAPE differently affects clinical outcome in two groups of patients stratified according to Ahlbäck stage (Stage II vs stage III).
However, it must be considered that in the present study this stratification would lead to a further decrease in sample size, so affecting statistical power. Nevertheless, evaluations of the impact of TAPE on clinical outcomes at different stages of the disease should be considered in future studies and this matter should be commented in discussion.
c) As reported in discussion (pag. 8 lines: 199-209), other cohort studies including patients with knee OA have previously reported significant improvement in pain and functional outcome after TAPE treatment. Therefore, in the discussion, the Authors should more strongly highlight the novel key results of the present study and how these results may introduce new aspects to be investigated in future research.
Reviewer 2 Report
Abstract:
Introduction
Line 39: change to joint disease
Line 73-74: Recommend just leaving safety and efficacy as purpose
Summary: How is periarticular embolization different from what is already reported in the literature? If it is not different, recommend keeping to published acronyms such as TAE of the knee joint. Additionally, I would recommend shortening the introduction and adding more relevant literature such as recently published Sham RCT and other prospective trials.
Materials and Methods
Is this a case series or retrospective study?
Why was the brachial artery selected? Is it routinely performed in the institution?
Are there other pictures that better demonstrate the tumor blush?
Results
Was MRI performed on all patients before and after procedure?
It is interesting that there were no procedural or access site related complications in any of the 17 patients.
Discussion
Line 210: these patients are not mentioned in the results. Pain is an adverse event and should be reported as such.
Were the patients use of analgesic medications recorded? Every study so far on GAE has reported analgesic use over time. Analgesic use is a major confounding factor.
This is one of the first papers where the benefits of GAE does not last past 1 month. There should be a more in-depth discussion as to why the authors hypothesize that GAE did not show benefits past 1 month – given that’s different from what’s been reported
The discussion should also mention the lack of post procedure imaging such as MRI to evaluate for the most feared complication which is bone infarcts – recently reported by Padia et al
Reviewer 3 Report
I would like to congratulate the authors for the study, but the paper should be reviewed by a native English speaker.
My comments are:
- In the introduction section, the authors must provide more bibliographical references. They only provide 9 references. The introduction should further justify the study.
- In the Method section, the authors must include point 2.1. Design. And they must modify the numbering of the following subsections
- Authors must include exclusion criteria
- The authors must include the references of the questionnaires that they used in the study.
- Authors should add a limitations section. They must put there the limitations found.
- The number of references is 22. The authors must provide more references to clarify the introduction and discussion section.
- The style of reference is correct.
Round 2
Reviewer 1 Report
No further modifications are needed